# Intrathymic Hemagioma: A Challenging Case Report with Special Focus on the Importance of Its Multidisciplinary Approach

**DOI:** 10.3390/pediatric17010013

**Published:** 2025-01-27

**Authors:** Milan Velimirovici, Anca Voichita Popoiu, Simona Cerbu, Calin Marius Popoiu, Florica Ramona Dorobantu, Borislav Dusan Caplar, Eugen Melnic, Anca Maria Cimpean, Larisa Cristina Tomescu, Maria Corina Stanciulescu

**Affiliations:** 1Doctoral School in Medicine, Victor Babes University of Medicine and Pharmacy, 300041 Timisoara, Romania; milan.velimirovici@umft.ro (M.V.); caplar.borislav@umft.ro (B.D.C.); tomescu.larisa@umft.ro (L.C.T.); 2Emergency Hospital for Children Louis Turcanu, 300041 Timisoara, Romania; popoiu.anca@umft.ro; 3Center of Expertise for Rare Vascular Disease in Children, Louis Turcanu Children Hospital, 300041 Timisoara, Romania; cerbu.simona@umft.ro (S.C.); mcpopoiu@umft.ro (C.M.P.); stanciulescu.maria@umft.ro (M.C.S.); 4Department XV of Orthopaedics, Traumatology, Urology and Medical Imaging, Discipline of Radiology and Medical Imaging, Victor Babes University of Medicine and Pharmacy, 300041 Timisoara, Romania; 5Department XI/Pediatric Surgery, Victor Babes University of Medicine and Pharmacy, 300041 Timisoara, Romania; 6Department of Neonatology, Faculty of Medicine and Pharmacy, University of Oradea, 410087 Oradea, Romania; 7Department of Prostheses Technology and Dental Materials, Faculty of Dental Medicine, Victor Babes University of Medicine and Pharmacy, 300041 Timisoara, Romania; 8Department of Pathology, Victor Babes University of Medicine and Pharmacy, 2004 Chișinău, Moldova; eugen.melnic@usmf.md; 9Department of Microscopic Morphology/Histology, Victor Babes University of Medicine and Pharmacy, 300041 Timisoara, Romania; 10Department of Obstetrics and Gynecology, Victor Babes University of Medicine and Pharmacy, 300041 Timisoara, Romania

**Keywords:** intrathymic hemangioma, endothelial cells, thymic vascular tumor

## Abstract

Mediastinal hemangiomas, particularly those of thymic origin, are rare phenomena. Due to its rarity, this pathologic condition is not characterized as related to the angiogenic profile of hemangioma endothelial cells. The diagnosis is challenging clinically and radiologically, and biopsies may not yield a definitive answer. Surgical resection offers the material for histologic diagnosis, relieves symptoms, and has a favorable long-term prognosis for such benign tumors. Sometimes, such benign tumors may have aggressive behavior and repeated recurrences but the causes responsible for this unpredictable evolution are not actually known. A case of intrathymic hemangioma diagnosed in a 16-year-old girl is presented here. We focused equally on a multidisciplinary approach to this challenging diagnosis but also on the characterization of the hemangioma endothelial cells profile not previously performed for such type of vascular anomalies. To define an antibodies panel for the evaluation of intrathymic hemangiomas may help in the full characterization of this rare vascular lesion, and subsequently focus on the new therapeutic targets which may be applied for cases with aggressive behavior.

## 1. Introduction

Thymus vascular tumors represent a group of rare vascular tumors and are currently only mentioned in the literature in 35 cases worldwide [1]. Most of them have been reported as “cavernous thymic hemangiomas” often diagnosed as a thymoma due to a partial overlapping of the clinical symptoms. Despite the fact that the first case of the thymic congenital hemangioma was reported in 1949 by a group of Spanish doctors [2], the diagnosis and immunophenotyping of thymic vascular tumors is still a real challenge for both clinicians and pathologists. The small number of cases reported in the literature most of them limited just to clinical and radiological findings [3,4,5,6] represented a disadvantage in establishing a well-specified immunohistochemical protocol to be widely applied in medical practice. 

WHO classification of Pediatric Tumors, 5th Ed, classifies thymus hemangiomas as vascular tumors of the thorax with the mesenchymal origin and unknown pathogenesis (https://tumourclassification.iarc.who.int/chaptercontent/35/191, accessed on 6 December 2024). More than 50% of thorax hemangiomas are asymptomatic but there are cases showing a high aggressiveness and poor prognosis. Classification of the thymus tumors does not include a particular subgroup related to thymic vascular lesions most probably due to their rarity.

For this reason, we considered immunophenotyping, the only case of vascular tumors with a location in the thymus that we identified in our workgroup, considering that thymic vascular lesion represents a real challenge.

## 2. Case Presentation

### 2.1. General Clinical Considerations

A 16-year-old female patient presented with a history of pain, muscle weakness, fatigue with minimal exertion, and dysphagia that had persisted for several months. Upon arrival at the neurologist’s office, the physician suggests conducting a test to measure the levels of acetylcholine antireceptor antibodies, which have been found to be elevated. She no longer attends his check-up appointments, but now he requires medication and chest compressions. She is quickly admitted, promptly intubated and transferred to the Medical Intensive Care Unit. Thorough investigations are crucial in establishing an accurate diagnosis for mediastinal tumors. When the overall state permits, it can be transferred to the Department of Paediatric Surgery for the necessary surgical procedure.

### 2.2. Radiological and CT Findings

A mediastinal mass was observed on the standard X-ray, prompting the recommendation for a CT scan with contrast to aid in distinguishing between teratoma, thymoma, and lymphoma at the initial evaluation. A tumor mass was detected during a CT examination in the right anterior mediastinal lodge. It measures 60/55 mm in the axial plane and 105 mm in the cranio-caudal plan. The tumour was found to be in direct contact with the pericardium, causing a slight displacement of the mediastinum to the left. Following the administration of the contrast substance, a strong contrast enhancement is detected, displaying a potentially uneven structure with areas of cystic necrosis. Figure 1a,b displays the CT examination imaging.

### 2.3. Surgical Management and Pathologic Evaluation

According to the CT scan findings, it was advised to proceed with surgery to remove the mediastinal mass. With confirmation of the formation by CT, it is decided the surgery for excision of the anterior mediastinal formation. The patient’s anesthetic risk was ASA III. We performed general anesthesia with sevoflurane pivotal ortho-tracheal intubation (IOT) without incidents. It is mounted: a central catheter in the right subclavian artery and urinary bladder tube. It was performed median sternotomy with the opening of the retrosternal space with sternal fragment haemostasias. In the anterior mediastinum in 1/3 upper, it was found a tumor mass with left and right extension of the midline from the ascending aorta to the upper left pulmonary artery in the thymus. It is practicing neat dissection. Excise the tumor formation together with the restant thymus. Suture with Prolen 5.0. Continue the procedure, and completely dissect the tumor with the opening of the pleura bilaterally. It is lavage with heated physiological saline. Check for haemostasis and anastomosis. Perform bilateral thoracic drainage. (Drain costo—bilateral diaphragmatic sinuses.) Wounds are restored on anatomical layers, followed by lung expansion. Continue with suction drainage (0.2 atm). Tumor and thymus formation should be taken for histopathological examination. The postoperative course was favorable. Mechanical ventilation was maintained for 24 h, and the patient was hemodynamically and respiratory stable. Bilateral pleural drainage is suppressed on the 5th day postoperative, then ultrasound and reduction of the amount of pericardial fluid is observed.

### 2.4. Macroscopic and Microscopic Histopathologic Evaluation

Upon surgical removal (Figure 2a–c), the tumor exhibited an irregular shape and measured 7.8/5.6/1.1 cm. Its color was observed to be gray-brown. The section exhibits a nodular, multichistic appearance and possesses an elastic consistency.

On histologic examination of the thymus biopsy, it was revealed a tumor mass which comprise over 50% of the section. The remaining thymic parenchyma showed relatively preserved architecture and organization but with two notable features: dilated septal vessels with focal stasis, and a predominance of degenerate Hassall corpuscles in the medullary of the remaining thymic parenchyma (Figure 3A,B).

The tumor observed under the microscope is lobulated and showed two distinct morphologic areas. One part consisted of tightly packed compact areas of tumor cells, while the other area consisted of vascular structures. These structures contained intraluminal red blood cells and were lined by endothelial cells (Figure 3C,D). Given the morphological heterogeneity and clinical outcome, it was necessary to conduct a differential diagnosis between a kaposiform hemangioendothelioma and a juvenile haemangioma. Given the unique nature of this case within our group and the rarity of the lesion globally, we decided it would be beneficial to seek a second opinion from an expert in the field. Based on his opinion and correlated to our previous observations we all agreed that the lesion is corresponding to a hemangioma usually with a benign evolution after surgery.

Considering the rarity of the case and the lack of extensive immunophenotyping reported in the literature for thymic hemangiomas, we found it necessary to conduct immunohistochemical reactions using a panel of less commonly used antibodies. This approach allowed us to identify specific immunohistochemical aspects that are valuable for diagnostic purposes and have potential prognostic implications

### 2.5. Immunohistochemistry

The antibody panel utilised consisted of various groups of markers. These markers included GLUT1 (Figure 4, ready to use mouse anti-human monoclonal antibodies, Leica, Biosystems, Newcastle Upon Tyne, UK), High Molecular Weight keratin 34 beta E12, HMW (ready to use, mouse anti-human monoclonal antibody, Agilent, Santa Clara, CA, USA) for verifying intrathymic localization, CD34 (ready to use, mouse anti-human monoclonal antibody, clone QBEnd 10, Agilent, Santa Clara, CA, USA) and F VIII (ready to use rabbit polyclonal antibody, Figure 5a,b) for identifying the endothelial origin of cells lining lumen-like structures, growth factors and corresponding receptors such as Platelet Derived Growth Factor B (PDGF_BB, Santa Cruz Biotechnology, Heidelberg, Germany) (lyophilized mouse anti-human monoclonal antibody, Agilent, Santa Clara, CA, USA) and its coresponding receptor PDGFRβ, Vascular Endothelial Growth Factor (VEGF-A, mouse anti-human monoclonal antibody, dilution 1:50, clone VG1, Agilent, Santa Clara, CA, USA) (Figure 5c). Confirmation of the tumor’s endothelial origin inside the thymus parenchyma was established through dual immunostaining using CD34/CK HMW (34 betaE12) (Figure 5d). Antibodies details together with details about detection and visualization methods are summarized in Table 1.

CV Mount (mounting medium from Leica Biosystems, Newcastle Ltd., Newcastle Upon Tyne, UK) was used as a permanent mounting medium for immunohistochemis-try-stained slides.

Distinct variations in marker expression and luminal morphology were noted between the compact and less compact regions of the vascular tumour. GLUT-1 was positive inside the endothelial cells of the hemagioma lesion (Figure 4).

Clusters of CK 34betaE12-positive epithelial cells were observed in compact areas, forming part of the CD34-positive tubular structures. Additionally, these cells were found attached to CD34-positive structures or interspersed among CD34-positive cells Cells positive for cytokeratin were observed in close proximity to vascular spaces in regions where the capillary pattern was most prevalent (Figure 5d).

Additionally, we assessed the markers for lymphatic endothelial activation and commitment (PROX1) and certified the lymphatic state (D2-40) (Figure 5e,f)

In compact areas, the tumor cell proliferation index showed a significant increase, with 50% of tumor cells actively proliferating. In contrast, the capillary-like areas had a much lower tumor cell proliferation rate, with no more than 10% of total tumor cells undergoing proliferation (Figure 5g,h).

One aspect that has received limited attention in the existing literature on hemangiomas, and has not been studied at all for thymic hemangiomas, is the expression of platelet-derived growth factor B and PDGFR beta. Positive expression of PDGF B was observed in both differentiated and undifferentiated areas of the hemangioma. The expression of the protein was limited to the endothelium that lines the capillary structures of the hemangioma (Figure 6a,b). The expression of PDGFR beta was found to be negative.

## 3. Discussion

Mediastinal hemangiomas have been continuously described since 1950 [7] but they still remain one of the less characterized lesions related to the endothelial cells phenotype or the existence of some particular types with unusual behavior. From about 600 mediastinal hemangiomas described in the literature [7], a number of 35 cases have been located inside the thymus [8]. Thymus is considered a rare location for hemangioma. Other rare and particular location of hemangioma were reported inside the ectopic thymus tissue with cervical location [9] but in older patients other than infants or adolescents. We presented here a rare vascular tumor with thymic location by combining clinical, radiological, microscopic and immunohistochemical methods diagnosed in a 16-year-old female patient. Usually, such lesions are extremely rare and are usually presented in the literature as having a cavernous morphology mostly [10,11,12] despite lacking this terminology (“cavernous”) in any classification of such lesions as the last version of WHO Classification of Pediatric Tumors and ISSVA classification. Present hemangioma of the thymus has a particular morphology, by having capillary areas intermixed with compact areas of endothelial cells showing a different phenotype related to proliferation and angiogenic growth factors expression. Other rare locations of infantile hemangioma outside the mediastinum were reported inside the uterine wall [13], or with intracranial [14], thyroid [15], synovial [16], hard palate [17], lower respiratory airways [18] or choroidal location [19].

VEGF and its corresponding receptors were previously characterized in the normal human thymus and thymus pathologic conditions other than hemangiomas [20]. VEGF was not reported to be expressed in thymic hemangioma before. In one of our previous studies, we found significant differences between gene expression profile of the VEGF pathway for proliferating versus involuting hemangiomas [21]. We reported there a divergent expression of 13 genes between proliferating versus involuting hemangiomas but none of them had the thymus as location.

PDGFs and their corresponding receptors were also studied by our team in the normal human thymus and thymomas, but not on thymus hemangioma [22]. During the evaluation of the present case, we found a high expression of PDGF B and PDGFR beta in the hemangioma area inside the thymus.

An important factor to consider is the role of thymic stromal cells in the development of thymic hemangioma. Simultaneous immunostaining The CD34/CK-HMW analysis revealed a reduction in the number of stromal epithelial cells that correlated with the growth of capillary structures in tumor development. The study found a significant decrease in the number of positive thymic stromal CK cells, which was directly related to the number of vascular structures in the hemangioma. The distribution, morphology, and interrelation with surrounding capillary structures of thymic stromal epithelial cells strongly indicated a process of trans-differentiation from thymic epithelial stromal cells into endothelial cells. These endothelial cells were able to narrow down the vascular structures within the thymic hemangioma.

The thymus is a very special and unique lymphoid organ with a unique histology of its stroma and molecular biology. It is highly challenging to discuss GLUT1 expression in the human thymus and its related disease due to GLUT1 heterogeneous expression in both stromal epithelial cells, endothelial cells from the thymus blood barrier and thymic lymphocytes [23]. Data about the GLUT1 expression in thymus hemangioma are practically inexistent in the literature but several papers describe GLUT1 expression inside the thymus and human thymomas. In the normal human thymus, GLUT1 has a lower expression compared to thymomas and thymic carcinomas [24], its expression having a highly predictive role in characterization of WHO thymomas subtypes [24], especially for differentiating between type B3 thymomas and thymic carcinomas [24]. However, some data support the increase of GLUT1 expression in type A thymoma as a predictive criterion for a high aggressivity of these lesions which are usually considered benign [25]. No data about intrathymic hemangioma progression/regression, age of appearance therapy response, or GLUT1 expression have been previously reported in the literature on adolescence.

Maybe the most unusual feature of the present intrathymic hemangioma, possibly shocking for some pathologists is its GLUT1 expression. Usually, GLUT1 is known to be positive in about 95 to 97% of infantile hemangiomas but it is a little bit exaggerated to classify our lesion as infantile hemangioma, due to its peculiarities which do not overlap with the criteria of infantile hemangioma lesion. Most of the infantile hemangiomas are located at the cutaneous level and the behavior of cutaneous infantile hemangioma related to its appearance age, proliferation and regression is done mostly by the direct observation of such hemangiomas located at the skin level. No data about the pathophysiology of intra-thymic hemangioma related to GLUT-1 expression have been found in the literature. Thus, we propose here two hypotheses which possibly may explain in part GLUT-1 expression for our case.

The data presented in our study provide evidence for the potential of stromal epithelial cells to transform into endothelial cells and actively contribute to the process of angiogenesis in these lesions. The positioning of HMW-positive CK cells in relation to capillary vessels, along with the presence of CD34 and HMW CK co-expression in the same cell, along with the morphological characteristics and the process of lumeinization, provide evidence for the involvement of thymic epithelial stromal cells in the development of thymic hemangiomas. Recently, Campinoti et al. [26], described a subpopulation of distinctive thymic stromal cells with a hybrid epithelial-mesenchymal phenotype able to reconstitute all cells of the human thymus stromal compartment. Our simple observation of CD34 and HMW CK co-expression may suggest the presence of epithelial-to-mesenchymal transition (EMT) as a potential mechanism of intra-thymic hemangioma development. EMT mechanism upregulates glucose transporters expression including GLUT-1 [27] Further research is necessary to fully understand the process of trans differentiation and the relationship between stromal epithelial cells and capillary vessels, given the infrequency of this lesion. The observational aspect of this research can be seen as a unique element.

Another possible mechanism for GLUT-1 expression may be induced by hypoxia related to hemangioma development. VEGF is also overexpressed during hypoxia induced by tumor growth as we observed for our lesion. Hypoxia-induced factors important for postnatal vasculogenesis (SDF-1α, MMP-9, VEGF-A, HIF-1α) are upregulated in children with proliferating hemangiomas. Other genes known to be expressed in hemangioma such as IGF-2 and GLUT-1 are also expressed in response to hypoxia [28].

## 4. Conclusions

We reported here a challenging GLUT-1 expressing intrathymic hemangioma in an adolescent. The hemangiomas located inside the thymus parenchyma are extremely rare lesions almost incompletely characterized related to their immunophenotype. Due to its rarity, multicenter studies will be needed to perform not just form the clinical point of view but mostly from microscopic and gene profiles with a special focus on its mechanism of development, and in some cases aggressiveness prognostic markers.

The present case does not fit to any vascular lesions described in the ISSVA classification or WHO Pediatric Tumors Classification with thymus location. Based on our clinical findings and immunohistochemical results we may assume that this thymic vascular lesion is closest as a diagnosis to a possible non-involuting hemangioma which, due to its location was not diagnosed and followed up till intrathoracic compressive symptoms appeared.

## Figures and Tables

**Figure 1 pediatrrep-17-00013-f001:**
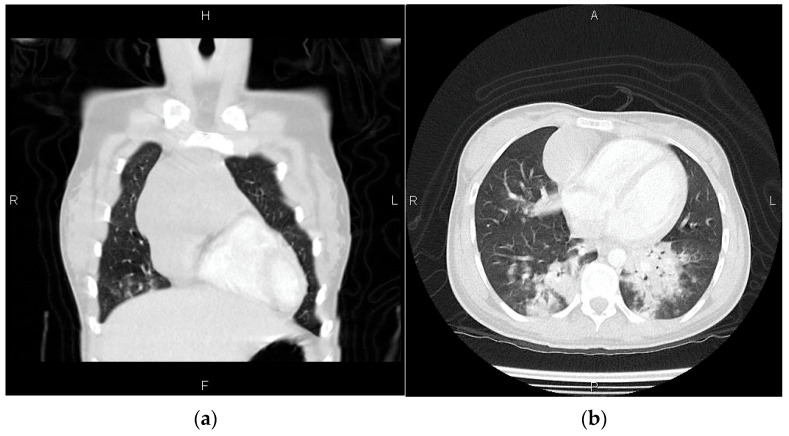
Within the anterior mediastinum, there is a solid vascularized process located in the thymic lodge (**a**). The tumor exerts pressure on the heart, causing displacement (**b**).

**Figure 2 pediatrrep-17-00013-f002:**
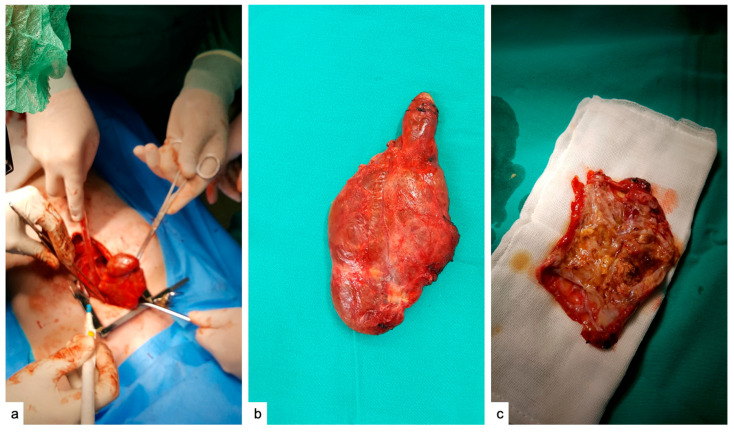
Surgical removal of the mediastinal tumor mass (**a**). Overview of the tumor before sectioning showed an oval tumor mass with a reddish color and with septae on the surface of the tumor which did not cause lobulation (**b**). Macroscopic view of the sectioned area was characterized by a multicystic appearance (**c**).

**Figure 3 pediatrrep-17-00013-f003:**
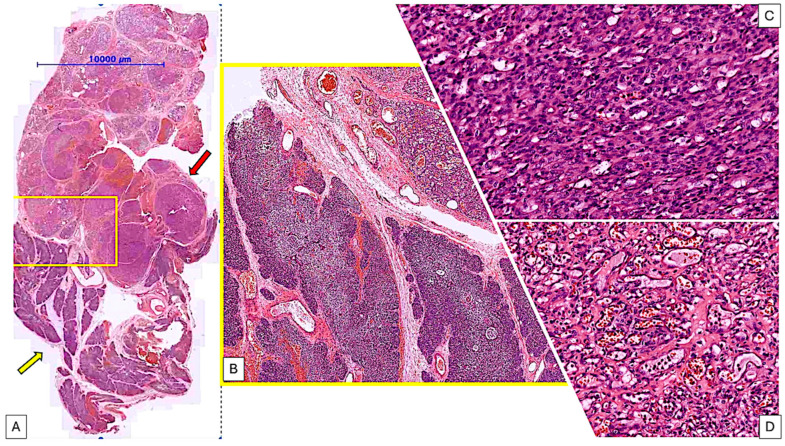
Overview (**A**) and details (**B**–**D**) on the tumor mass from the thymus. Restant normal thymus tissue may be observed on the yellow arrow from (**A**) while tumor mass was detected on the red arrow. Yellow quadran from (**A**) showed the edge in between normal and tumor tissue and may be visualized in detail in (**B**). Two morphologically different aspects of thymus lesion were detected, displaying the characteristics of a capillary haemangioma, which are defined as solid areas with different patterns of vascular structures: lobular growth pattern with slightly larger vessels with an evident lumen, containing red blood cells (**D**) and a smaller and more dense vessels, with no evident lumens (**C**).

**Figure 4 pediatrrep-17-00013-f004:**
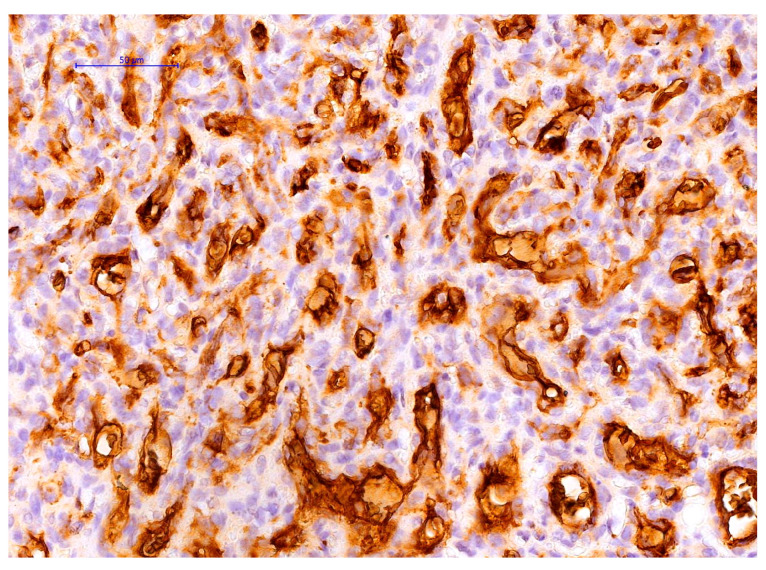
GLUT1 expression on endothelial cells from our case.

**Figure 5 pediatrrep-17-00013-f005:**
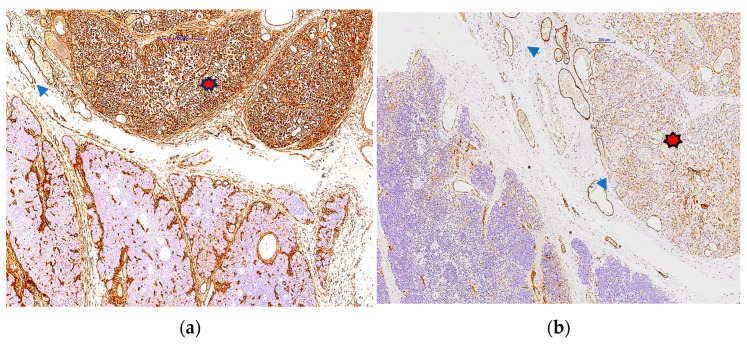
Immunophenotyping of thymus hemangioma. CD34 was intensely positive in the hemangioma are (**a**, red star) compared to adjacent normal thymus parenchyma. F VIII-related antigen was also positive in the hemangioma area (**b**, red star) but with low intensity and inconstant expression (**b**). VEGF A was intensely positive in the hemangioma area while in the normal thymus parenchyma, its expression was restricted to subcapsular stromal epithelial cells and Hassall’ s corpuscles (**c**). Double immunostaining with HMW CK 34 beta E12 (red) and CD34 (brown) highlighted the decrease of keratin-positive epithelial cells density in the hemangioma area ((**d**), blue star) where scattered stained in red thymic epithelial cells were found closely attached to capillaries ((**d**), arrowheads). PROX1 was found positive in the endothelial cells from vessels adjacent to the hemangioma but not inside the hemangioma lesion (**e**, red star). Podoplanin (clone D2-40) expression mostly overlaps with the PROX1 expression highlighting an increase in lymphatic microvessel density around the hemangioma (**f**) but not inside the hemangioma (**f**, red star).Low proliferative Ki67 index for the capillary areas of hemangioma (**g**, yellow star) compared to high proliferation found in the compact area of the hemangiomas (**h**, yellow star).

**Figure 6 pediatrrep-17-00013-f006:**
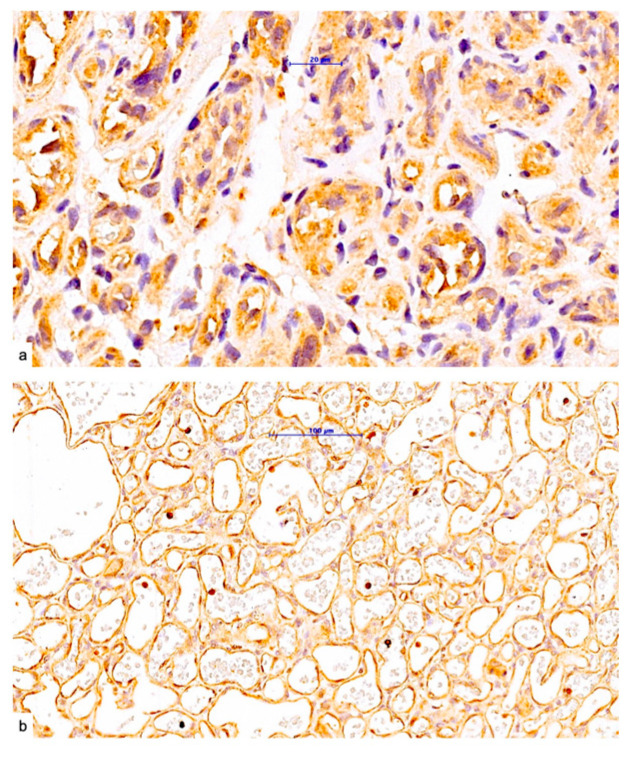
Expression of PDGF B in less differentiated (**a**) and well-differentiated (**b**) thymic hemangiomas areas.

**Table 1 pediatrrep-17-00013-t001:** Primary antibodies and visualization systems used for the present case immunohistochemistry.

Antibody Name	Source	Clone	Dilution	Detection/Visualization System	Antigen Unmasking	Incubation Time with Primary Antibody
CD34	Leica Biosystem Newcastle Ltd., Newcastle Upon Tyne, UK	QBEnd 10	Prediluted	Bond Polymer Refine Detection System (Leica Biosystems, Newcastle Upon Tyne, UK), 15 min	Bond Epitope Retrieval Solution 2, (Leica Biosystems, Newcastle Upon Tyne, UK)—20 min	30 min, room temperature
VEGF	ReliaTech GmbH, Lindener Str. 15, 38300 Wolfenbuttel, Germany	VG1	Liofilised 1:50	Bond Polymer Refine Detection System (Leica Biosystems, Newcastle Upon Tyne, UK), 15 min	Bond Epitope Retrieval Solution 2, (Leica Biosystems, Newcastle Upon Tyne, UK)—20 min	30 min, room temperature
FVIII	Leica Biosystem Newcastle Ltd., Newcastle Upon Tyne, UK	36B11	Prediluted	Bond Polymer Refine Detection System (Leica Biosystems, Newcastle Upon Tyne, UK), 15 min	Bond Epitope Retrieval Solution 2, (Leica Biosystems, Newcastle Upon Tyne, UK)—20 min	30 min, room temperature
PDGF-A	Santa Cruz Biotechnology, www.scbt.com, accessed on December 6, 2024	Polyclonal N30	1:25	Bond Polymer Refine Detection System (Leica Biosystems, Newcastle Upon Tyne, UK), 15 min	Bond Epitope Retrieval Solution 2, (Leica Biosystems, Newcastle Upon Tyne, UK)—20 min	30 min, room temperature
PDGF B	Santa Cruz Biotechnology, www.scbt.com, accessed on December 6, 2024	Monoclonal, F3	1:100	Bond Polymer Refine Detection System (Leica Biosystems, Newcastle Upon Tyne, UK), 15 min	Bond Epitope Retrieval Solution 2, (Leica Biosystems, Newcastle Upon Tyne, UK)—20 min	30 min, room temperature
PDGFR beta	Santa Cruz Biotechnology, www.scbt.com, accessed on December 6, 2024	Polyclonal P20, SC -339	1:30	Bond Polymer Refine Detection System (Leica Biosystems, Newcastle Upon Tyne, UK), 15 min	Bond Epitope Retrieval Solution 2, (Leica Biosystems, Newcastle Upon Tyne, UK)—20 min	30 min, room temperature
Ki 67	Leica Biosystem Newcastle Ltd., Newcastle Upon Tyne, UK	MM1	Prediluted	Bond Polymer Refine Detection System (Leica Biosystems, Newcastle Upon Tyne, UK), 15 min	Bond Epitope Retrieval Solution 2, (Leica Biosystems, Newcastle Upon Tyne, UK)—20 min	30 min, room temperature
Prox 1	ReliaTech GmbH, Lindener Str. 15, 38300 Wolfenbuttel, Germany	polyclonal	Liofilized 1: 10	Bond Polymer Refine Detection System (Leica Biosystems, Newcastle Upon Tyne, UK), 15 min	Bond Epitope Retrieval Solution 2, (Leica Biosystems, Newcastle Upon Tyne, UK)—20 min	60 min, room temperature
Podoplanin	Abcam plc, Discovery Drive, Cambridge Biomedical Campus, Cambridge, CB2 0AX, UK	gp36	1:40	Bond Polymer Refine Detection System (Leica Biosystems, Newcastle Upon Tyne, UK), 15 min	Bond Epitope Retrieval Solution 2, (Leica Biosystems, Newcastle Upon Tyne, UK)—20 min	30 min, room temperature
GLUT 1	Abcam plc, Discovery Drive, Cambridge Biomedical Campus, Cambridge, CB2 0AX, UK	Monoclonal, EPR3915	1:250	Bond Polymer Refine Detection System (Leica Biosystems, Newcastle Upon Tyne, UK), 15 min	Bond Epitope Retrieval Solution 2, (Leica Biosystems, Newcastle Upon Tyne, UK)—20 min	30 min, room temperature
High Molecular Weight Keratin 34beta E12	Leica Biosystem Newcastle Ltd., Newcastle Upon Tyne, UK	34BETA E12	Prediluted	Bond Polymer Refine Detection System (Leica Biosystems, Newcastle Upon Tyne, UK), 15 min	Bond Enzyme 1 Leica Biosystem Newcastle Ltd., Newcastle Upon Tyne, UK, 5 min	30 min, room temperature

## Data Availability

All data related to this manuscript are available upon request to Anca Maria Cimpean personal database.

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
