# Peer review of "Intrathymic Hemagioma: A Challenging Case Report with Special Focus on the Importance of Its Multidisciplinary Approach"

_pediatrrep, 2025, doi:10.3390/pediatric17010013_

Round 1
Reviewer 1 Report
Comments and Suggestions for Authors
Introduction
The authors described a rare case of thymic hemangioma.
Authors should refer to the WHO 5th Ed classification (Paediatric Tumours)/ISSVA Classification 2018, also in introduction.
2.Case presentation
2.1. General clinical considerations.
2.3. Surgical management and pathologic evaluation.
- approximately 7.8/5.6/1.1 cm 7.8x5.6x1.1 cm. Why "approximately", they are precise measurements.
- "under a microscope" I personally prefer "on histological examination".
- "One part consisted of tightly packed compact areas of tumour cells, while the other area consisted of vascular structures."
I think that both areas are formed by vessels: in figure 3C they are smaller and more dense, with no evident lumens, in figure 3D they are slightly larger and with an evident lumen, containing red blood cells.
I would suggest describing the characteristics of the recurrence (clinical picture, radiographic picture, further interventions, further histological examinations, current state).
2.4. Immunohistochemistry.
I would suggest introducing a table with the antibodies used, including also Ki67.
There is an error in the indication of figures 4a,b,c, etc. because they are figures 5 a,b,c, etc.
Discussion
The term infantile hemangioma is introduced in the discussion when until then it had only been referred to as hemangioma. This can create confusion (208-210).
Various punctuation errors detected.
Author Response
RESPONSE TO REVIEWER 1
Dear Reviewer 1,
Thank you for your feedback related to our manuscript. Your observations and suggestions were highy valuable for us and we tried to address all of them in a positive manner. Please, find them highlighted in red in the manuscript body but also in the present response.
Kind regards,
AMCimpean
Introduction
The authors described a rare case of thymic hemangioma.
Authors should refer to the WHO 5th Ed classification (Paediatric Tumours)/ISSVA Classification 2018, also in introduction.
We inserted a paragraph. Please let me know if you have any suggestion to improve this paragraph. Thank you!
2.Case presentation
2.1. General clinical considerations.
2.3. Surgical management and pathologic evaluation.
- approximately 7.8/5.6/1.1 cm 7.8x5.6x1.1 cm. Why "approximately", they are precise measurements.
Thank you for your observation. We deleted “approximately”!
- "under a microscope" I personally prefer "on histological examination".
We re-phrase the sentence. Thanks!
- "One part consisted of tightly packed compact areas of tumour cells, while the other area consisted of vascular structures."
I think that both areas are formed by vessels: in figure 3C they are smaller and more dense, with no evident lumens, in figure 3D they are slightly larger and with an evident lumen, containing red blood cells.
Pertinent observation, we made changes, thanks!
I would suggest describing the characteristics of the recurrence (clinical picture, radiographic picture, further interventions, further histological examinations, current state).
Unfortunately we had no pictures from recurrence due to the fact that when the patient came back was a real EMERGENCY and needed rapid surgical intervention!. We do really apologize we are not able to address your request but we consider that what we presented is enough to be credible for such a rare case.
2.4. Immunohistochemistry.
I would suggest introducing a table with the antibodies used, including also Ki67.
We added this table!
There is an error in the indication of figures 4a,b,c, etc. because they are figures 5 a,b,c, etc.
You are right! We corrected these errors!
Discussion
The term infantile hemangioma is introduced in the discussion when until then it had only been referred to as hemangioma. This can create confusion (208-210).
We removed the term infantile.
Various punctuation errors detected.
We corrected them with Quillbot assistant!

Reviewer 2 Report
Comments and Suggestions for Authors
The paper is aimed to present a challenging case of intrathymic hemangioma in 16-year-old girl.
The authors support their decision to publish the case by the rarity of the condition and some unique features of this particular case. They briefly overview the problem of intramediastinal hemangiomas, describe clinical and radiological data and diagnostic journey of the patient. Surgical procedure described in details and supplied with the demonstrative pictures.
The most important data presented and discussed are related to the histology and immunohystochemistry findings. The authors present their data, step by step unveiling the findings and opinions of different specialists and final consensus decision. The most interesting finding discovered and discussed is related to the GLUT-1 expression which is highly unusual for this type of matured tumours and was shocking for the pathologists, as the authors emotionally wrote.
The authors give some hypothetical explanations for this phenomenon and propose the following steps to the studies.
The paper is inspired by the rare but clinically and scientifically important case. The authors meticulously documented the findings and discussed the results. It is even more important and interesting that they present the consequential opinions of pathologists including international experts.
Some text editing is recommended in terms of formal English.
Author Response
RESPONSE TO REVIEWER 2
Dear Reviewer 2,
Thank you for your feedback related to our manuscript. Your observations and suggestions were highy valuable for us and we tried to address all of them in a positive manner. Please, find them highlighted in red in the manuscript body but also in the present response.
Kind regards,
AMCimpean
The paper is aimed to present a challenging case of intrathymic hemangioma in 16-year-old girl.
The authors support their decision to publish the case by the rarity of the condition and some unique features of this particular case. They briefly overview the problem of intramediastinal hemangiomas, describe clinical and radiological data and diagnostic journey of the patient. Surgical procedure described in details and supplied with the demonstrative pictures.
Thank you for the appreciation!
The most important data presented and discussed are related to the histology and immunohystochemistry findings. The authors present their data, step by step unveiling the findings and opinions of different specialists and final consensus decision. The most interesting finding discovered and discussed is related to the GLUT-1 expression which is highly unusual for this type of matured tumours and was shocking for the pathologists, as the authors emotionally wrote.
Related to GLUT1 expression we discussed this unusual expression and we gave data from the literature that GLUT1 may be expressed also in adult hemangiomas with other location as renal or adrenal location.
The authors give some hypothetical explanations for this phenomenon and propose the following steps to the studies.
The paper is inspired by the rare but clinically and scientifically important case. The authors meticulously documented the findings and discussed the results. It is even more important and interesting that they present the consequential opinions of pathologists including international experts.
Thank you for the appreciation!
Some text editing is recommended in terms of formal English.
There were corrected with Quillbot assistant! Thank you!

Reviewer 3 Report
Comments and Suggestions for Authors
Dear Authors,
Vascular anomalies are an extremely vast field, much debated in the specialized literature lately. Despite the many publications in the last 10 years (52,145 results on a simple search on PubMed), this pathology is still a mystery both from the pathological and immunohistochemical point of view, as well as the malformative associations and clinical manifestations. I therefore consider that those rare cases, correctly diagnosed and pathologically classified, must be debated and presented.
Although the case presented by you is a very interesting one, the manuscript requires a major revision before it can be published. Therefore, I would have some suggestions and recommendations.
The title: you refer in the title to the importance of the multidisciplinary approach; I totally agree with its necessity, but in your manuscript you refer almost exclusively to the pathological aspects. The consultation and opinion of the geneticist, the cardiologist, the anesthetist and even the pediatric surgeon is missing: what were the difficulties of the surgical intervention? what surgical approach was used? what was the evolution of the patient?
Abstract: in the Abstract text that appear in the system you used the term "vascular malformation". Please revise and correct: if it is an hemangioma, than it is not a vascular malformation. Please refer to ISSVA classification.
Introduction: The introduction is poor and lacks adequate references. Even if the specialized literature presents only 35 cases (or 39 as you mentioned in the Discussions chapter?), the lesion you want to present must be introduced in the general context of vascular anomalies, respectively vascular tumors, and if considered, infantile hemangiomas .
Case presentation: What I specified initially (related to the title of the manuscript) must be detailed in this section. A case presentation cannot be limited only to the pathological aspects of the excised tumor.
Related to the management of this condition: have you considered other types of treatment, such as corticosteroid therapy, immunosuppression therapy, treatment with Propranolol (given the suspicion of infantile hemangioma) or treatment with Sirolimus? If not, I would suggest discussing these issues in the Discussion section.
Line 118-123: With all due respect to Professor Saul Suster, I believe that this paragraph has no point in the case presentation. Mr. Professor can be cited for his studies and can be mentioned in the Acknowledgments section.
Line 123-124: What do you mean by " our case was particular due to its recurrence"?
Discussion: In this chapter, I believe that the case itself, the clinical manifestations, the challenges of medical and surgical treatment, the framing and correct classification of the lesion and the comparison with similar cases published in the specialized literature should be discussed first. All the more so as the authors recognize the fact that the diagnosis of infantile hemangioma is a forced one, the lesion not fitting into the classic aspect described in the ISSVA classification.
The entire manuscript must be revised for the English language and typographical errors.
Comments on the Quality of English Language
The manuscript has to be revised by a native English speaker.
Author Response
RESPONSE TO REVIEWER 3
Dear Reviewer 3,
Thank you for your feedback related to our manuscript. Your observations and suggestions were highy valuable for us and we tried to address all of them in a positive manner. Please, find them highlighted in red in the manuscript body but also in the present response.
Kind regards,
AMCimpean
Dear Authors,
Vascular anomalies are an extremely vast field, much debated in the specialized literature lately. Despite the many publications in the last 10 years (52,145 results on a simple search on PubMed), this pathology is still a mystery both from the pathological and immunohistochemical point of view, as well as the malformative associations and clinical manifestations. I therefore consider that those rare cases, correctly diagnosed and pathologically classified, must be debated and presented.
We are totally agree with you.
Although the case presented by you is a very interesting one, the manuscript requires a major revision before it can be published. Therefore, I would have some suggestions and recommendations.
Thank you for all suggestions. They highly contributed to the improvement of our paper.
The title: you refer in the title to the importance of the multidisciplinary approach; I totally agree with its necessity, but in your manuscript you refer almost exclusively to the pathological aspects. The consultation and opinion of the geneticist, the cardiologist, the anesthetist and even the pediatric surgeon is missing: what were the difficulties of the surgical intervention? what surgical approach was used? what was the evolution of the patient?
We added a paragraph related to anesthesia and surgical procedures and postoperative evolution of the patient.
Abstract: in the Abstract text that appear in the system you used the term "vascular malformation".
The abstract from the system was not updated due to the fact that after full submission we are not able to change the abstract in the system. But in the manuscript, the abstract does not contain any vascular malformation terms. Thank you but in the system is not our fault.
Please revise and correct: if it is a hemangioma, than it is not a vascular malformation. Please refer to ISSVA classification.
Introduction: The introduction is poor and lacks adequate references. Even if the specialized literature presents only 35 cases (or 39 as you mentioned in the Discussions chapter?), the lesion you want to present must be introduced in the general context of vascular anomalies, respectively vascular tumors, and if considered, infantile hemangiomas .
We do no really know what we can do!. Most of these cases are published as cavernous hemangiomas , a term which is not found in the ISSVA classificationa nymore, now. SO, at our previous submission to another journal one of the reviewers adviced us to cut all this references based on this issue. We are in a big dillemma. Please let us know if it would be possible to keep our introduction in this form due to the fact that is a case presentation not a research paper. Thank you!
Case presentation: What I specified initially (related to the title of the manuscript) must be detailed in this section. A case presentation cannot be limited only to the pathological aspects of the excised tumor.
Dear Reviewer 3, with all my respect for your time and effort to revise our manuscript, I would like to mention that our case presentation is focused on pathological and immunohistochemical faindings but is not limited to this part. At the beginning of Case presentation section you will find a brief clinical context description of the case in the subsection 2.1. I will insert also here
- General clinical considerations.
A 16-year-old female patient presented with a history of pain, muscle weakness, fatigue with minimal exertion, and dysphagia that had persisted for several months. Upon arrival at the neurologist's office, the physician suggests conducting a test to measure the levels of acetylcholine antireceptor antibodies, which have been found to be elevated. She no longer attends his check-up appointments, but now he requires medication and chest compressions. She is quickly admitted, promptly intubated, and transferred to the Medical Intensive Care Unit. Thorough investigations are crucial in establishing an accurate diagnosis for mediastinal tumours. When the overall state permits, it can be transferred to the Department of Paediatric Surgery for the necessary surgical procedure.
Sub-section 2.1detailed above, involved a pediatrician, a pediatric surgeon and a pediatric anestesiologist. So, it was a multidisciplinary team.
After this subsection we presented radiological and CT findings in the subsection 2.2 and these come to complete multidisciplinarity. There are different medical speciality working together to this case and thus it is multidisciplinary approach. Last, but not least, we focused to pathology and immunohistochemistry by adding a pathologist in the team and this was crucial due to the fact that if I remember well the mediastinum is called Pandorra ‘s Box and the pathologist must stated type of the tumor from heterogeneous mesenchymal tumors of the mediastinum or other more aggressive type as lymphomas and so on…Pathology findings explained also multidisciplinar approach. Usually such cases already published are resumed to one to three pictures of CT findings and brief surgical description but not to an extensive combined histopathological and immunohistochemical approach as we did. Last, but not least one of the strenghts of our case presentation related to multidisciplinarity is that the extensive Immunohistochemical approach is part of a more comprehensive research activity from one of my PhD student related to hemangiomas. There are ery rare case presentations where research data are intermixed with an usual case presentation. We did it here due to the fact that as you asked in the next paragraph, several therapies are conditioned by these analysis.
Related to the management of this condition: have you considered other types of treatment, such as corticosteroid therapy, immunosuppression therapy, treatment with Propranolol (given the suspicion of infantile hemangioma) or treatment with Sirolimus? If not, I would suggest discussing these issues in the Discussion section.
Line 118-123: With all due respect to Professor Saul Suster, I believe that this paragraph has no point in the case presentation. Mr. Professor can be cited for his studies and can be mentioned in the Acknowledgments section.
We removed the paragraph and re-phrased it without names. Thank you!
Line 123-124: What do you mean by " our case was particular due to its recurrence"?
Discussion: In this chapter, I believe that the case itself, the clinical manifestations, the challenges of medical and surgical treatment, the framing and correct classification of the lesion and the comparison with similar cases published in the specialized literature should be discussed first.
We did not find in the literature similar cases. The only ones we found were almost all cavernous hemangiomas (te term is not now in ISSVA classification anymore, but in all literature this term is extensively used!!!! So, in this field is a big.big, very big vonfudion related to the type of thymus hemangioma. We suspected that it is a non involuting congenital hemangioma (not yet described in the literature at the level of the thymus)not yet characterized related to its evolution and recurrence. That ‘s why we consider interesting to publish this due to its controversial behaviour. It would be a challenging case and might be the trigger for publishing other similar cases. I consdier this a new entity with a particualr behaviour due to its location in a special microenvironment as thymus.
So we apologize but we have no other similar cases to comapre with. Thus we are not able to make discussions as you suggested. Please let us know if you have case suggestions from the literature which we may use for this request from you. Thanks in advance!
All the more so as the authors recognize the fact that the diagnosis of infantile hemangioma is a forced one, the lesion not fitting into the classic aspect described in the ISSVA classification.
Yes you are right, this case does not fit with a ny ISSVA classification, nor to WHO classification of Pediatric tumors , 5th edition. Thus we assume that this case can be, based on IHC stainings and histopathology a congenital non involuting infantile hemangioma with a particular location in the thymus.
The entire manuscript must be revised for the English language and typographical errors.
We managed this and corrected most of them by using Quillbot Package.
With all my best regards,
AMCimpean

Round 2
Reviewer 3 Report
Comments and Suggestions for Authors
Dear Authors,
Thank you for your responses. I consider that the case is interesting and the manuscript can be published.
Best Regards.